# Preclinical Enzyme Replacement Therapy with a Recombinant β-Galactosidase-Lectin Fusion for CNS Delivery and Treatment of GM1-Gangliosidosis

**DOI:** 10.3390/cells11162579

**Published:** 2022-08-19

**Authors:** Jason Andrew Weesner, Ida Annunziata, Tianhong Yang, Walter Acosta, Elida Gomero, Huimin Hu, Diantha van de Vlekkert, Jorge Ayala, Xiaohui Qiu, Leigh Ellen Fremuth, David N. Radin, Carole L. Cramer, Alessandra d’Azzo

**Affiliations:** 1Department of Genetics, St. Jude Children’s Research Hospital, Memphis, TN 38105, USA; 2Department of Anatomy and Physiology, University of Tennessee Health Science Center, Memphis, TN 38163, USA; 3Compliance Office, St. Jude Children’s Research Hospital, Memphis, TN 38105, USA; 4BioStrategies, LC, P.O. Box 2428, State University, Jonesboro, AR 72467, USA

**Keywords:** lysosomal storage disease, GM1, ERT, mβ-Gal:RTB, CNS

## Abstract

GM1-gangliosidosis is a catastrophic, neurodegenerative lysosomal storage disease caused by a deficiency of lysosomal β-galactosidase (β-Gal). The primary substrate of the enzyme is GM1-ganglioside (GM1), a sialylated glycosphingolipid abundant in nervous tissue. Patients with GM1-gangliosidosis present with massive and progressive accumulation of GM1 in the central nervous system (CNS), which leads to mental and motor decline, progressive neurodegeneration, and early death. No therapy is currently available for this lysosomal storage disease. Here, we describe a proof-of-concept preclinical study toward the development of enzyme replacement therapy (ERT) for GM1-gangliosidosis using a recombinant murine β-Gal fused to the plant lectin subunit B of ricin (mβ-Gal:RTB). We show that long-term, bi-weekly systemic injection of mβ-Gal:RTB in the *β-Gal*^−/−^ mouse model resulted in widespread internalization of the enzyme by cells of visceral organs, with consequent restoration of enzyme activity. Most importantly, β-Gal activity was detected in several brain regions. This was accompanied by a reduction of accumulated GM1, reversal of neuroinflammation, and decrease in the apoptotic marker caspase 3. These results indicate that the RTB lectin delivery module enhances both the CNS-biodistribution pattern and the therapeutic efficacy of the β-Gal ERT, with the potential to translate to a clinical setting for the treatment of GM1-gangliosidosis.

## 1. Introduction

Lysosomal storage diseases (LSDs) comprise the largest group of monogenic neurodegenerative disorders in children, with a collective incidence of 1:5000 livebirths [1]. Many LSDs are caused by a single deficiency of lysosomal hydrolases, mostly glycosidases. Soluble enzymes are synthesized as high molecular weight (MW) precursors and acquire their mature and active state within the acidic lysosomal compartment [2,3,4,5]. A portion of the enzyme precursors is also secreted by default with the bulk of secretory proteins, but it maintains the capacity to be reinternalized via receptor-mediated endocytosis by neighboring or distant cells [4,6]. This mechanism of “cross-correction,” which is unique for soluble lysosomal enzymes, has been extensively exploited for the development of diverse therapeutic approaches, including enzyme replacement therapy (ERT) [7,8]. ERT is considered the least invasive and safest therapy for LSDs, although the high costs associated with the manufacturing of recombinant enzymes make this approach less accessible in a clinical setting, especially for rare LSDs. The biggest limitation of ERT is the incapacity of the recombinant enzymes to reach the central nervous system (CNS) and effectively cross the blood–brain and blood–cerebral spinal fluid (CSF) barriers [9,10]. To overcome these caveats, investigators have tested the in situ delivery of recombinant lysosomal enzymes via intrathecal or intracerebroventricular injections in preclinical animal models of LSDs; however, translating this approach into the patient population should be exercised with caution, considering the debilitating status of LSD patients [10,11]. More recently, several studies in LSD models have focused on the development of modified fusion proteins, which have the potential to aid in the transport of recombinant enzyme(s) injected intravenously (IV) to the CNS. These include the fusion of lysosomal enzymes (i.e., β-galactosidase (β-GAL), iduronate-2-sulfatase, and sulfamidase) to genetically engineered monoclonal antibody against the human transferrin receptor or the human insulin receptor for the treatment of mouse models of GM1-gangliosidosis, mucopolysaccharidosis type II (MPSII) and MPSIIIA [12,13,14]. Another modification is based on the fusion of the non-toxic subunit B of ricin (RTB) to lysosomal iduronidase [15] for the treatment of MPSI or β-GAL [16], which has been tested in this study for the treatment of GM1-gangliosidosis. As a lectin, RTB binds to diverse and abundant surface glycoproteins (including receptors) and glycolipids to direct endocytosis, lysosomal delivery and transcytosis. In contrast to other lysosomal enzyme delivery strategies, RTB-mediated uptake is not dependent upon the presence or abundance of a specific receptor for efficient delivery of its cargo enzyme into target cells, resulting in distinct pharmacokinetics, low cell uptake saturability, and access to “hard-to-treat” tissues, including the CNS [17].

Lysosomal β-GAL catalyzes the hydrolysis of terminal β-linked galactose residues from glycoconjugates. The enzyme has a high affinity for two substrates, GM1-ganglioside (GM1), a sialylated glycosphingolipid (GSL) abundant in the nervous system, and proteoglycan, keratan sulfate [18,19]. Deficiency of β-GAL is at the basis of two LSDs: GM1-gangliosidosis, a GSL storage disease, and Morquio disease type B, a MPSIVB [18,19]. GM1-gangliosidosis patients present with a continuum of disease severity that primarily affects the nervous system. They are usually classified into three groups: infantile, late infantile and juvenile/adult, based on age of onset of the systemic clinical manifestations, and cognitive/neurological delays [18,19]. Although there is no clear genotype–phenotype correlation, mutations linked to the infantile form of the disease mostly fall within the catalytic core of the enzyme, while mutations associated with the late infantile and juvenile/adult forms are present on the protein surface. The latter mutations are usually found in patients with higher residual enzyme activity [18,19].

β-GAL is part of a lysosomal multienzyme complex (LMC), which includes two other lysosomal enzymes, the sialidase, neuraminidase 1 (NEU1) and the carboxypeptidase, protective protein/cathepsin A (PPCA) [20]. The association of PPCA with NEU1 and β-GAL in humans is pivotal for catalytic activation and stability of the two glycosidases in lysosomes [20,21]. β-GAL maintains its activity outside the LMC, although the enzyme is less stable in the absence of a functional PPCA, while NEU1 is catalytically inert when not in complex with PPCA [20]. In addition, studies performed in an animal model [7,22] and cell lines of galactosialidosis [23], caused by a primary defect of PPCA, have revealed that murine β-Gal is less dependent on its association with PPCA for its stability [7,23].

Five murine models that phenocopy GM1-gangliosidosis have been created to date; they carry mutations in different exons of the *Glb1* gene and were generated using different targeting strategies [24,25,26,27,28]. Although they differ in lifespan and time of onset of clinical and biochemical abnormalities, they all share a progressive accumulation of GM1 in the nervous system and present similar phenotypic characteristics. The mouse model of GM1-gangliosidosis used in this study (*β-Gal*^−/−^) closely resembles the early onset form of the disease [25]. Similar to patients, *β-Gal*^−/−^ mice develop a widespread nervous system pathology, presenting with tremors, ataxia and abnormal gait, which culminates in rigidity and paralysis of the hind limbs and reduced lifespan [25]. The brain and spinal cord of mutant mice show a massive, age-dependent accumulation of GM1 and its asialo-derivative GA1 [25], resulting in neuronal cell death and progressive astrogliosis and microgliosis [29]. This model has been extensively used to investigate the mechanisms of pathogenesis that may explain the features of GM1-gangliosidosis in children [25,30,31,32]. It was found that the progressive build-up of GM1 in *β-Gal*^−/−^ brains leads to the redistribution of this ganglioside in neuronal intracellular membranes, including those of the endoplasmic reticulum (ER), which induces depletion of the ER calcium (Ca^2+^) store and consequent activation of the unfolded protein response (UPR)-mediated cell death pathway [32]. It was further demonstrated that GM1 accumulates at specific sites of apposition between the ER and mitochondrial membranes, the so-called mitochondria-associated ER membranes (MAMs). This favors the formation of these contact sites and facilitates the transfer of Ca^2+^ from the ER to the mitochondria, ultimately evoking the mitochondria apoptotic pathway [30]. *β-Gal*^−/−^ mice have also been exploited for testing various therapeutic modalities, including in vivo and ex vivo gene therapy, with promising improvement of the CNS pathology [31,33,34,35].

Here, we sought to experiment with a minimally invasive ERT approach, producing a recombinant β-Gal protein fused to the RTB in the leaves of *Nicotiana benthamiana* plants [16]. This novel lysosomal delivery approach supports distinct in vivo biodistribution and has the potential to reach hard-to-treat organs, such as the brain [16,17]. A recombinant fusion protein composed of RTB and human β-Gal (hβ-Gal:RTB) has been previously shown to maintain all the biochemical characteristics of the endogenous enzyme and to restore β-Gal activity when used in vitro to correct GM1-gangliosidosis fibroblasts [16]. Considering the stability of the murine β-Gal outside the LMC, we have now designed a proof-of-concept preclinical protocol to test whether RTB fused with murine β-Gal (mβ-Gal:RTB) can correct the neuropathological features of *β-Gal*^−/−^ mice. We demonstrate that the systemic long-term administration of mβ-Gal:RTB increased β-Gal activity in all visceral organs. Most importantly, the recombinant enzyme was detected in the CNS, resulting in reduced GM1 levels and the reversal of neuronal cell death and neuroinflammation. This positive outcome holds promise for the use of RTB-β-Gal as a novel ERT approach for the treatment of GM1-gangliosidosis in the clinic.

## 2. Materials and Methods

### 2.1. Animals and Ethics

Animals were housed in a fully Association for Assessment and Accreditation of Laboratory Animal Care (AAALAC)-accredited animal facility with controlled temperature (22 °C), humidity and lighting (alternating 12-h light/dark cycles). Food and water were provided ad libitum. All procedures in the mice were performed according to animal protocols approved by the St Jude Children’s Research Hospital Institutional Animal Care and Use Committee (IACUC) and the National Institutes of Health (NIH) guidelines. All experiments were performed in *β-Gal*^−/−^ [25] mice and WT littermate controls in the C57BL/6 background at 4 weeks and 10 weeks of age.

### 2.2. Cloning and Expression of the Murine RTB Fusion Construct

The mβ-Gal:RTB vector was constructed similar to the hβ-Gal:RTB as described previously [16]. To construct the mβ-Gal:RTB expressing vector, we used tobacco-codon optimized murine *Glb1* (*mGlb1*, GenBank accession no.: AH001860.2) with its native signal peptide replaced with an optimized signal peptide sequence derived from patatin tuber storage protein (*PoSP*). The *PoSP:mGlb1^opt^* gene was synthesized (GeneArt^®^, Piscataway, NJ, USA; Life Technologies, Waltham, MA, USA), fused to *RTB* at its 3′-end and cloned into a pBC-SK(-) vector. The *PoSP:mGlb1^opt^:RTB* cassette was further subcloned into the plant expression vector derived from the pBIB-Kan binary vector and transformed into *Agrobacterium tumefaciens* strain LBA4404 by electroporation [16].

### 2.3. Production and Purification of mβ-Gal:RTB

The mβ-Gal:RTB used in this study was manufactured at BioStrategies, LC. To produce mβ-Gal:RTB, *Agrobacterium* cultures harboring a *mGlb1^opt^:RTB* expression construct were infiltrated into the leaves of *Nicotiana benthamiana* plants by vacuum infiltration, as described previously [16,36]. Leaves were harvested at 4 days post-infiltration and stored at −80 °C.

To purify mβ-Gal:RTB, the frozen infiltrated leaf material was homogenized in 100 mM Tris, 150 mM MgCl_2,_ 10 mM N_2_S_2_O_5,_ 20 mM D-(+)-galactose, pH 7.5 in a commercial blender (Waring). After centrifugation at 16,000× *g* for 20 min at 4 °C, the supernatant was filtered through three layers of Miracloth (Calbiochem) and further clarified by adding 6.1 mM of CaCl_2_ and 6.1 mM of phytic acid (Sigma, St. Louis, MO, USA), followed by centrifugation at 15,000× *g* for 20 min at 4 °C. Proteins in the supernatant were precipitated with 47% ammonium sulfate saturation for 30 min at 4 °C followed by centrifugation at 12,500× *g* for 20 min at 4 °C. The mβ-Gal:RTB-containing pellet was then resuspended in resuspension buffer (RB) (100 mM Tris, 150 mM NaCl_2_ pH 7.8) for 30 min at 4 °C. After further clarification by centrifugation at 20,000× *g* for 20 min at 4 °C, the crude extracts were filtered through 0.8, 0.45, and 0.2-micron filters before affinity purification.

Affinity chromatography was performed in an MT20 column (Bio-Rad, Hercules, CA, USA) packed with α-lactose agarose gel (EY Laboratories, San Mateo, CA, USA) in an AKTA pure FPLC system (GE Healthcare, Chicago, IL, USA). The lactose affinity column was equilibrated with 5 column volumes (CV) of RB. Crude extracts were loaded at 10 mL/min and washed with 4 CV of RB. The mβ-Gal:RTB fusions were eluted from the lactose column with elution buffer (200 mM Tris, 150 mM NaCl_2_, 300 mM of D-(+)-galactose, pH 7.8).

Elution fractions from the lactose column were concentrated using a 30K VivaSpin Column (Sartorious, Göttingen, Germany) prior to size exclusion chromatography (SEC). The SEC column (HiLoad 26/60 Superdex 200 pg; GE Healthcare) was equilibrated with 2 CV equilibration buffer (50 mM sodium phosphate, 150 mM NaCl_2_, pH 7.6) before injecting the samples. Protein was eluted with the same buffer at 1.0 mL/min. The peak corresponding to the protein of interest was concentrated and buffer exchanged to DPBS buffer (Sigma, St. Louis, MO, USA) using 30K VivaSpin columns. The final product was formulated in DPBS containing 0.4 M trehalose (MP Biomedicals, Irvine, CA, USA) and 0.2 mg/mL of Polysorbate 80 (Sigma). Protein concentrations were determined by measuring absorbance at A280 (MW, 115 kDa/Ext. Coeff 184,580 ∈ M-1 cm-1) using Cytation 3 with a Take3™ multi-volume plate (BioTek, Santa Clara, CA, USA). Purity (>95%) was confirmed by Coomassie gel. Endotoxin levels were certified to be <1 EU per µg of protein by the LAL method (Endosafe^®^ nexgen-PTS™, Charles River, Wilminton, MA, USA). Aliquoted samples were stored at −80 °C.

### 2.4. In Vivo Treatment with mβ-Gal:RTB

For 24 h dose-dependent injections, 4-week-old *β-Gal*^−/−^ mice were first injected with cyproheptadine HCL (CPH) intraperitonially (IP) to reduce an inflammatory response to ERT injections [37]. After 15–30 min, the mice were injected intravenously (IV) with mβ-Gal:RTB with a concentration of either 3 mg β-Gal equivalents/kg or 5 mg β-Gal equivalents/kg of mouse weight. 24 h post-injection, injected mice, WT mice, and uninjected *β-Gal*^−/−^ littermates, were sacrificed, and tissues were collected, without perfusion, and divided, with half stored in formalin for IHC analysis. The other half was snap frozen in liquid N_2_. For long term injections, *β-Gal*^−/−^ mice were injected twice a week for 6 weeks starting at 4 weeks old. Mice were injected IP with CPH for 15 min, followed by IV injections of mβ-Gal:RTB with a concentration of 5 μg/g of mouse weight. 24 h after the last injection, mice were sacrificed by CO_2_ and tissues were collected in formalin or frozen in liquid N_2_.

### 2.5. Western Blot Analyses

Frozen tissue samples were homogenized in cold HPLC-grade water using Tissue Lyser II (Qiagen, Hilden, Germany) with 5 mm metal beads. For immunoblotting, protein concentrations were determined in homogenized tissues using a Pierce BCA kit (Thermo Scientific, Waltham, MA, USA). Proteins were separated by sodium dodecyl sulfate–polyacrylamide gel electrophoresis on precast Mini-PROTEAN 4–20% TGX gels (Bio-Rad, Hercules, CA, USA) under reducing conditions and transferred to a polyvinylidene difluoride (PVDF) membrane (Millipore, Burlington, MA, USA). Membranes were incubated for 1 h in blocking buffer (5% dry milk in tris-buffered saline (TBS)-Tween-20) at room temperature (RT) and subsequently probed with 1:100 anti-murine or anti-human β-Gal (produced in house) or 1:1000 anti-Caspase 3 (Cell Signaling, Danvers, MA, USA, Cat. #9661S) antibodies diluted in 5% bovine serum albumin (BSA)/TBS-Tween-20 overnight at 4 °C under agitation. Following this step, blots were washed 3X with TBS-Tween-20 for 5 min and incubated with 1:10,000 anti-rabbit HRP secondary antibody (Jackson Immunoresearch, West Grove, PA, USA, Cat. #705-035-152) for 1 h at RT. Blots were washed 3X with TBS-Tween-20 for 5 min and developed using the Clarity Max™ Western ECL Substrate (Bio-Rad, Hercules, CA, USA).

### 2.6. Enzyme Activity Assays

For β-Gal enzyme activity, tissue lysates were diluted and 10 µL of each sample was transferred to a 96-well clear bottom black plate and incubated with 20 µL of β-Gal substrate (1 mM 4-methylumbelliferyl-β-D-galactopyranoside (Sigma, St. Louis, MO, USA), 100 mM Na-Acetate pH 4.3, 100 mM NaCl). For Neu1 activity, 5 µL of each sample was incubated with 5 µL of synthetic Neu1 substrate (4-methylumbelliferyl-α-D-N-acetylneuraminic acid (Sigma, St. Louis, MO, USA), 100 mM Na-Acetate pH 4.3, 100 mM NaCl). Fluorescent standards (40 µM 4-methylumbeliferone, 0.9% NaCl, 0.02% Na-Azide) were added and plates were incubated for 1 h at 37 °C. Reactions were stopped by adding 200 µL of 0.5 M carbonate buffer, pH 10.7, to each well. The fluorescence was measured on a plate reader (EX-355, EM-460). The net fluorescence values were compared with those of the linear fluorescent standard curve and used to calculate specific enzyme activities. Activities were calculated as nanomoles of substrate converted per hour per milligram of protein (nmol/h/mg).

### 2.7. Immunohistochemistry and IF

Tissues were fixed in 10% neutral-buffered formalin for 48 h at RT and processed, sectioned and stained with a standard H&E for overall morphologic assessment.

For immunohistochemistry (IHC), 6-μm-thick paraffin sections were subjected to deparaffinization and antigen retrieval in 10 mM sodium citrate/0.1% Tween-20 using pressure cooker methods. After permeabilization for 45 min in 0.5% Triton™ X-100/PBS and blocking for 1 h in blocking buffer (2% BSA/0.5% Triton™ X-100/PBS), sections were incubated overnight at 4 °C with 1:100 anti-murine β-gal (produced in-house) or 1:500 anti-Iba1 (Wako, Osaka, Japan, Cat. #019-19741) antibodies diluted in blocking buffer. Sections were then washed 3X for 5 min in blocking buffer and subsequently incubated with 1:400 anti-rabbit IgG biotin conjugated secondary antibody (Jackson ImmunoResearch, West Grove, PA, USA, Cat. #111-065-003) diluted in blocking buffer for 2 h at RT. Endogenous peroxidase was quenched by incubating the sections with 5% hydrogen peroxidase/PBS for 15 min at RT. Antibody detection was performed using the VECTASTAIN Elite ABC Kit (Vector Laboratories, Newark, CA, USA, Cat. No. PK-6100) and diaminobenzidine substrate (Invitrogen, Waltham, MA, USA, Cat. No. 750118) and sections were counterstained with hematoxylin according to standard methods.

For immunofluorescence (IF), sections (6-μm) were cut and subjected to deparaffinization. Antigen retrieval was performed in 10 mM sodium citrate/0.1% Tween-20 buffer using the pressure cooker method. The sections were permeabilized in 0.5% Triton™ X-100/PBS for 45 min and blocked in Blocking Buffer (2% BSA/0.5% Triton™ X-100/PBS) for 1 h at RT. Slides were incubated with 1:500 anti-Iba1 (Wako, Osaka, Japan, Cat. #019-19741) or 1:1500 anti-GFAP (Dako, Santa Clara, CA, USA, Cat. #Z0334) antibodies in blocking buffer overnight at 4 °C. The next day, sections were washed 3X for 5 min in blocking buffer and subsequently incubated with 1:400 anti-rabbit IgG biotin conjugated secondary antibody (Jackson ImmunoResearch, West Grove, PA, USA, Cat. #111-065-003) in blocking buffer for 2 h at RT. Sections were washed twice for 5 min in blocking buffer and then washed once with PBS for 10 min at RT and mounted using ProLong™ Gold antifade containing DAPI (Invitrogen, Waltham, MA, USA). Images were taken using a Lionheart FX automated microscope (BioTek, Santa Clara, CA, USA) with a 20× objective.

### 2.8. HPTLC Analysis of GM1

To extract the lipids from brain tissues, 150 µL of tissue lysates were used, and an 8× volume of methanol was added to each sample (8:3 volume CH_3_OH:dH_2_O) and samples were vortexed at RT. Chloroform equaling half the volume used for methanol was added to each sample (4:8:3 volume CHCl_3_:CH_3_OH:dH_2_O) and vortexed at RT. The mixture was then centrifuged at 1200× *g* for 15 min at RT and the supernatant was measured (µL) and transferred into a fresh Eppendorf tube. Water (0.173 times the volume) was added, and the samples were vortexed and centrifuged at 1200× *g* for 15 min at RT. The upper polar phase was collected in a new tube and evaporated at 48 °C overnight until the lipid pellet remained.

To analyze GM1 content by high-performance thin-layer chromatography (HPTLC), lipid pellets were resuspended in 20–50 µL of 4:8:3, CHCl_3_:CH_3_OH:dH_2_O to a final concentration of 100 μg/μL protein for WT and 30 μg/μL for treated and untreated *β-Gal*^−/−^ samples. Then, 5 µL of the sample was loaded to a 10 × 20 cm silicone coated thin-layer chromatography (TLC) plate (Millipore, Burlington, MA, USA) using the Automatic TLC Sampler ATS 4 (Camag). GM1 (1 µg/µL) isolated from bovine brain (Sigma, St. Louis, MO, USA, Cat no. G7641) in 4:8:3 CHCl_3_:CH_3_OH:dH_2_O was plated as a standard. Plates were developed in an Automatic Developing Chamber ADC2 (Camag, Muttenz, Switzerland) using 60:35:8 CHCl_3_:CH_3_OH:0.25% KCL as the development solution. Plates were then sprayed with resolving solution (2% resorcinol, 80% HCL, 5% 0.1 M copper sulfate) and heated on a hot plate for 30 min at 95 °C until bands appeared. Plates were imaged on the TLC Visualizer 2 (Camag, Muttenz, Switzerland) and bands containing GM1 were quantified using the GM1 standard.

### 2.9. GM1 Sandwich and IgG1 ELISA

The GM1 levels in the cortex, cerebellum, brain stem and hippocampus homogenates from WT, *β-Gal*^−/−^ and treated *β-Gal*^−/−^ mice were measured by enzyme-linked immunosorbent assay (ELISA). Briefly, immulon^®^ 4HBX 96-well plates (Thermo Scientific, Waltham, MA, USA) were coated with the Cholera toxin B subunit for GM1 capture. After binding of GM1 content on lysate samples, GM1 was detected using anti-GM1 antibody (Abcam, Cambridge, UK, Cat# ab23943). The GM1 levels from samples were determined based on a standard curve of commercial monosialoganglioside-GM1 (Sigma, St. Louis, MO, USA, Cat# G7641) and the final concentration was normalized to nanogram of GM1 per microgram of total soluble proteins (ng of GM1/µg total soluble protein).

IgG1 serum concentration against the RTB-fusion product and RTB domain were measured by ELISA using the SBA Clonotyping System-C57BL/6-HRP (Southern Biotech, Birmingham, AL, USA, Cat#: 5300-05B) following the manufactured instructions. Briefly, plates were coated overnight with 2.5 μg/mL of mβ-Gal:RTB or RTB domain produced and purified in an analogous system. Serum samples were incubated at a 1:400 dilution for 1 h. Measured IgG1 levels were quantified using an IgG1 standard curve (Southern Biotech, Birmingham, AL, USA, Cat#: 5300-01).

### 2.10. Statistical Analysis

Statistical analyses were performed with Student’s *t* test, ordinary one-way analysis of variance (ANOVA) or the Brown-Forsyth and Welch ANOVA test using GraphPad Prism. Quantitative data are presented as mean ± standard deviation (SD). Statistical *p* values of <0.05 were considered significant.

## 3. Results

### 3.1. Characterization of Murine β-Gal Fused to RTB

We first assessed the properties of mβ-Gal:RTB in a series of biochemical and cellular uptake experiments. On immunoblots, purified mβ-Gal:RTB showed the 120 kDa RTB-conjugated β-Gal precursor band using a murine-specific anti-β-Gal antibody (Figure 1A). In addition, we showed that a band of 85 kDa, corresponding to the unconjugated precursor form of β-Gal was visible (Figure 1A). High expression levels of the purified recombinant protein were accompanied by sustained β-Gal activity (1.14 × 10^6^ nmol/h/mg) (Figure 1B). Lastly, we performed an in vitro uptake assay using purified mβ-Gal:RTB administered to the culture medium of murine embryonic fibroblasts (MEFs) isolated from *β-Gal*^−/−^ mice, and skin fibroblasts derived from a mRNA negative GM1-gangliosidosis patient [38,39]. In both deficient cells, mβ-Gal:RTB restored the enzyme activity to values that exceeded those of the control cells (Figure 1C). The proteolytic processing of mβ-Gal:RTB after uptake in deficient cells was monitored in total cell lysates immunoblotted and probed with an anti-β-Gal antibody. The 84 kDa precursor β-Gal was normally processed into its mature 64/20 kDa two-chain enzyme (Figure 1D,E).

Overall, these results show that the mβ-Gal:RTB produced in plants yields an enzyme that is internalized and processed by cells and was used for further in vivo analyses in *β-Gal*^−/−^ mice.

### 3.2. β-Gal^−^^/−^ Mice Treated with mβ-Gal:RTB Show Increased β-Gal Activity and Biodistribution in Visceral Organs

We next tested whether the RTB delivery system could provide sufficient β-Gal activity to key sites of GM1 accumulation and mitigate the cascade of downstream pathologies characteristic of the disease in the *β-Gal*^−/−^ mouse model. To determine the most suitable dosage for long-term treatment with mβ-Gal:RTB, three *β-Gal*^−/−^ mice were first injected at postnatal day 28 (±2 days) with two concentrations of the recombinant enzyme, 3- or 5-mg/kg, and harvested 24 h post-injection. IV injection of both doses resulted in a significant increase in activity in all the tested visceral organs compared to *β-Gal*^−/−^ non-injected animals (Figure 2A–D). In the liver and spleen of injected mice, even the low dose of the recombinant enzyme restored β-Gal activity to WT levels (Figure 2A,B). In the lung and kidney, organs were generally more difficult to correct, and both doses substantially increased β-Gal activity over the values in *β-Gal*^−/−^ mice, but the highest dose had the most prominent effect (Figure 2C,D).

It is noteworthy that several *β-Gal*^−/−^ tissues had between 1–10% residual enzyme activity, as previously reported [25]. This activity towards 4-methylumbelliferyl β-galactoside substrate could be due to other lysosomal enzymes, i.e., galactosidases, which are normally expressed in the *β-Ga*l^−/−^ mice. These levels are low and considered background, as these mice have been previously shown to be mRNA negative [25].

Remarkably, comparing the effects of the two doses on several regions of the brain 24 h post-injection showed a more robust increase in β-Gal activity in the cortex and cerebellum of mice treated with the high dose of mβ-Gal:RTB (Figure 2E–H). Based on these overall results, a long-term study was performed in a larger cohort of *β-Gal*^−/−^ mice (11 mice) to assess the distribution and therapeutic efficacy of mβ-Gal:RTB at a dose of 5 mg/kg. *β-Gal*^−/−^ mice were injected bi-weekly for 6 consecutive weeks. Following treatment, an increase in β-Gal activity was measured in all visceral organs compared to the untreated *β-Gal*^−/−^ mice (Figure 3A–D). β-Gal activity was restored to normal or higher values in the liver and spleen of treated mice (Figure 3A,B), while in lungs and kidneys, enzyme values reached 12–14% of WT levels (Figure 3C,D). Given the potential influence of β-Gal deficiency on Neu1 lysosomal levels [11], we also measured Neu1 activity in treated mice. Neu1 activity, which was significantly increased in *β-Gal*^−/−^ liver, was restored to WT levels following treatment with mβ-Gal:RTB (Appendix A). A similar trend was observed in the spleen and kidney, while no changes in Neu1 activity were measured in the lungs (Appendix A).

To assess whether mβ-Gal:RTB would be effectively taken up and processed in the lysosomes of treated mice, we performed immunoblot analyses of tissue homogenates from visceral organs. All treated organs showed the presence of mβ-Gal:RTB fusion protein, indicating that it was efficiently internalized and processed into its mature 64/20 kDa form in lysosomes (Figure 3E–H, Appendix A). Importantly, in all visceral organs of the treated mice, we detected a substantial amount of the processed mature 64 kDa form of β-Gal, while trace amounts of the 20 kDa chain were seen only in tissues with the highest β-Gal levels, such as the liver and spleen (Figure 3E–L, Appendix A).

To determine the distribution of mβ-Gal:RTB following the 6-week-long treatment, tissue sections from the liver, spleen and kidney were analyzed for the presence of β-Gal using immunohistochemistry. In agreement with the results of the Western blot analyses, β-Gal was detected in discrete lysosomal puncta in hepatocytes, splenocytes and epithelial cells of the kidney distal tubules in all treated mice, indicating that the recombinant enzyme was effectively internalized by cells of different organs (Figure 4).

### 3.3. Treatment with mβ-Gal:RTB Led to Increased β-Gal Activity and Decreased GM1 Levels in the Brain

To prove that long-term treatment of *β-Gal*^−/−^ mice with mβ-Gal:RTB led to delivery of the therapeutic enzyme to the CNS, different brain regions and spinal cord tissues were analyzed for the presence of the recombinant enzyme. In all brain regions of the injected mice, we detected a significant increase in enzyme activity over the values measured in non-injected *β-Gal*^−/−^ littermates (Figure 5A–D). Western blots of the same brain regions showed the presence of the mature 64 kDa β-Gal chain (Figure 5E–L, Appendix A). Neu1 activity in the cortex showed a similar trend to that seen in the visceral organs; Neu1 activity was increased in *β-Gal*^−/−^ samples and returned to WT levels following treatment (Appendix A). In contrast, Neu1 activity was decreased in the cerebellum, brain stem and spinal cord of *β-Gal*^−/−^ mice and showed no change in these tissues of the treated animals (Appendix A).

We next tested the downstream effects of restored β-Gal activity by measuring the levels of GM1 in the cerebral cortex, cerebellum, brain stem and spinal cord of mβ-Gal:RTB-treated animals. A substantial reduction of GM1 was observed in the cortex (34%), cerebellum (38%), brain stem (43%) and spinal cord (28%) when assessed using HPTLC (Figure 6A–H and Appendix A). Similar results were obtained using a sandwich ELISA with the cortex, showing 33% reduction (KO = 20.33 ng GM1, mβ-Gal:RTB = 13.47 ng GM1), cerebellum 38% (KO = 8.84 ng GM1, mβ-Gal:RTB = 5.42 ng GM1), brain stem 25% (KO = 11.25 ng GM1, mβ-Gal:RTB = 8.44 ng GM1), and spinal cord 15% (KO = 5.31 ng GM1, mβ-Gal:RTB = 4.52 ng GM1) (Figure 6I–L). As a readout for the total ganglioside content, we also measured the total bound sialic acid for each sample and observed a similar trend to that seen for GM1 (Appendix A). The increase in the sialic acid content in *β-Gal*^−/−^ mice paralleled the increase in GM1, but it was clearly reduced after treatment with mβ-Gal:RTB (Appendix A). Taken together, these analyses demonstrate that the dose of mβ-Gal:RTB chosen for in vivo treatment was sufficient to promote GM1 degradation in the CNS tissue of *β-Gal*^−/−^ mice.

To determine if *β-Gal*^−/−^ mice elicit an immune response to either the RTB domain or the recombinant mβ-Gal:RTB fusion protein, we performed an IgG1 ELISA with sera collected from mβ-Gal:RTB treated *β-Gal*^−/−^ mice. Antibodies against the mβ-Gal:RTB recombinant fusion protein were detected in all animals, whereas only 3 mice raised antibodies against the RTB domain at very low levels, suggesting that the antibodies were mainly directed against β-gal epitopes (Appendix A). We further analyzed the effect of antibody levels on the efficacy of the drug by correlating the levels of anti-drug IgG1s present in the sera to the reduced GM1 levels in the treated mice (Appendix A). We found no direct correlation between the IgG1 levels directed to mβ-Gal:RTB and the reduction of GM1 in the cortex. The 3 mice with the highest measured IgG1 levels against the recombinant protein still showed an average of 50% GM1 reduction, which was more than the combined average of 45% reduction, indicating that these antibodies were non-neutralizing and did not affect the efficacy of the therapeutic outcome. These results are also in line with other findings using other RTB-fusion proteins, where the development of anti-drug antibodies was shown not to affect the delivery of the enzyme to the CNS [15,17,40].

### 3.4. Amelioration of Phenotypic Abnormalities in mβ-Gal:RTB-Treated β-Gal^−/−^ Brain

In H&E-stained brain sections, the presence of numerous, expanded lysosomes in neurons, filled with storage, is one of the overt phenotypic abnormalities of *β-Gal*^−/−^ mice. To assess whether treatment of these mice with mβ-Gal:RTB resulted in amelioration of brain histopathology, brain sections were stained with H&E and evaluated morphologically. In all brain regions of the treated mice, including the thalamus, one of the most affected areas, there was a clear reduction in lysosomal vacuolation (Figure 7), indicating that the recombinant enzyme was taken up by affected neurons and cleared some of the lysosomal storage.

We next wanted to assess whether reduction of GM1 accumulation would rescue, at least in part, the neuronal cell death observed in the *β-Gal*^−/−^ mice [30,32]. For this purpose, we assessed the levels of the canonical apoptotic marker caspase 3 in the cortex of treated animals [41]. *β-Gal*^−/−^ mice showed a 4-fold increase in caspase 3 levels when compared to the control animals (Appendix A). In contrast, caspase 3 levels were reduced by more than 50% following treatment with mβ-Gal:RTB (Appendix A).

Neuronal apoptosis in the *β-Gal*^−/−^ mice is known to elicit a neuroinflammatory response characterized by prominent astrogliosis and microgliosis [31]. Brains from mβ-Gal:RTB-injected and non-injected mice were stained with glial fibrillary acidic protein (GFAP), a marker of astrocytes, and ionized Ca^2+^ binding adaptor molecule 1 (Iba1), a marker of reactive microglia. The injected mice showed a significant reduction in the number of reactive astrocytes and microglia throughout the entire brain, including the thalamus (Figure 8 and Figure 9). Quantification of microgliosis and astrogliosis in mβ-Gal:RTB-injected mice showed a statistically significant 3.6-fold reduction of Iba1-positive microglia and a 4.3-fold reduction of GFAP-positive astrocytes (Appendix A). Furthermore, mβ-Gal:RTB treatment resulted in changes in the size and shape of microglia, which are parameters indicative of their activated status (Figure 10A–C and Appendix A). Untreated *β-Gal*^−/−^ mice showed reactive, activated, amoeboid microglia with retracted processes and small branching, while mβ-Gal:RTB-injected mice showed normalized, ramified microglia with extensive branching similar to those seen in control WT brains (Figure 10A,B and Appendix A). Taken together, these results reiterate that the chosen dose of mβ-Gal:RTB was sufficient to revert significant phenotypic alterations downstream of GM1 accumulation in the brain.

## 4. Discussion

A major challenge in the treatment of LSDs arises from the complexity and diversity of their clinical phenotypes and prominent CNS involvement. Delivery of therapeutics to the brain has proven difficult due to the selective permeability of the blood–brain and blood–CSF barriers [9]. However, various therapeutic approaches have been tested preclinically in small and large animal models of LSDs, some of which have led to the development of clinical trials for these diseases. Specifically, the *β-Gal*^−/−^ mice have been extensively exploited to test the efficacy of various therapies, such as substrate deprivation therapy, as well as ex vivo and in vivo gene therapy [31,33,34,35,42]. In all instances, correction of the visceral organ pathology was accompanied by partial reversal of some of the neurodegenerative aspects of this disease, including neuronal cell death and neuroinflammation [18]. The therapeutic outcome of in vivo adeno-associated virus (AAV) mediated-gene therapy in the *β-Gal*^−/−^ mouse model used in this study has been successfully translated in a phase 1/2 clinical trial [43] aimed at assessing the safety and efficacy of a single-dose gene transfer vector AAV9/human-GLB1 by intravenous infusion. However, this clinical trial is currently on hold, limiting the therapeutic options currently available for GM1 gangliosidosis patients.

To date, ERT remains the most widely used and the only FDA-approved therapeutic approach for the treatment of LSDs. Early clinical trials using ERT were developed in the 1990s for Gaucher patients using purified enzymes from the human placenta [44]. Since then, ERT has been used in many clinical trials for the treatment of Fabry disease, Pompe disease, neuronal ceroid lipofuscinosis, alpha-mannosidosis, acid lipase deficiency, and many mucopolysaccharidoses (MPS I, II, IVA, VI and VII) [45,46,47,48,49,50,51,52,53,54]. The use of this approach over a period spanning ~30 years has demonstrated its safety and highlighted its efficacy for the treatment of non-neuropathic patients. The procedure is minimally invasive; the enzyme is administered by IV injection, but the costs of mass production of a clinical-grade recombinant product for long-term patient care are high since this therapy requires recurrent infusion of the deficient enzymes [9]. ERT has also proven limited in its ability to correct the full range of clinical manifestations characteristic of some LSDs. For example, in the case of Pompe disease, an ERT approach was successful in reverting the cardiac pathology but did not correct skeletal muscle disease, and patients treated with this approach showed persistent muscle degeneration [45].

The most relevant limitation of ERT is the inability of the recombinant enzyme to reach the CNS, making this therapeutic approach unsuitable for treating the neurological aspects of LSDs [9]. In the case of the *β-Gal*^−/−^ mouse model, the only way to target the brain via ERT has been by direct intracerebroventricular administration of recombinant β-GAL, resulting in broad biodistribution of the enzyme to affected areas of the brain, substantial reduction of GM1 levels and reversal of neuropathology [11]. However, this approach would be difficult to readily translate to the clinic for the treatment of patients. In another ERT study aimed to deliver the enzyme to the brain by IV injection, the recombinant human β-GAL was fused to the heavy chain of a mouse monoclonal antibody against the murine transferrin receptor [13]. However, this approach only slightly improved motor function, was not able to correct all visceral organs, and, most importantly, did not show the presence of the enzyme in the brain [13].

The results of the current study suggest that these limitations may be overcome by the incorporation of the RTB delivery module to facilitate more effective access of the recombinant enzyme via ERT to reach critical regions of pathology, especially in the CNS. This delivery-enhanced approach has allowed us to target therapeutic β-Gal not only to the visceral organs of *β-Gal*^−/−^ mice, but most importantly the brain, leading to a reversion of key neurodegenerative abnormalities of GM1-gangliosidosis, including GM1 accumulation, neuronal cell death, and neuroinflammation. It appears that the mβ-Gal:RTB product is broadly distributed throughout the CNS (e.g., see Figure 4), similar to the biodistributions seen with thalamic infusion of an AAV9 vector expression human β-GAL [33] or following direct intrathecal or intracerebroventricular injections of a recombinant human enzyme [11]. In addition, the amelioration of downstream biomarkers of GM1 pathology (neuroinflammation, astrogliosis, microgliosis) suggests that the bioactive product is delivered to critical sites of GM1-induced pathology, requiring not only mobilization across the BBB but also trafficking across the brain parenchyma. Thus, this approach could potentially become the therapy of choice for neuropathic GM1 gangliosidosis patients, especially if treatment begins in the early stages of the disease.

## Figures and Tables

**Figure 1 cells-11-02579-f001:**
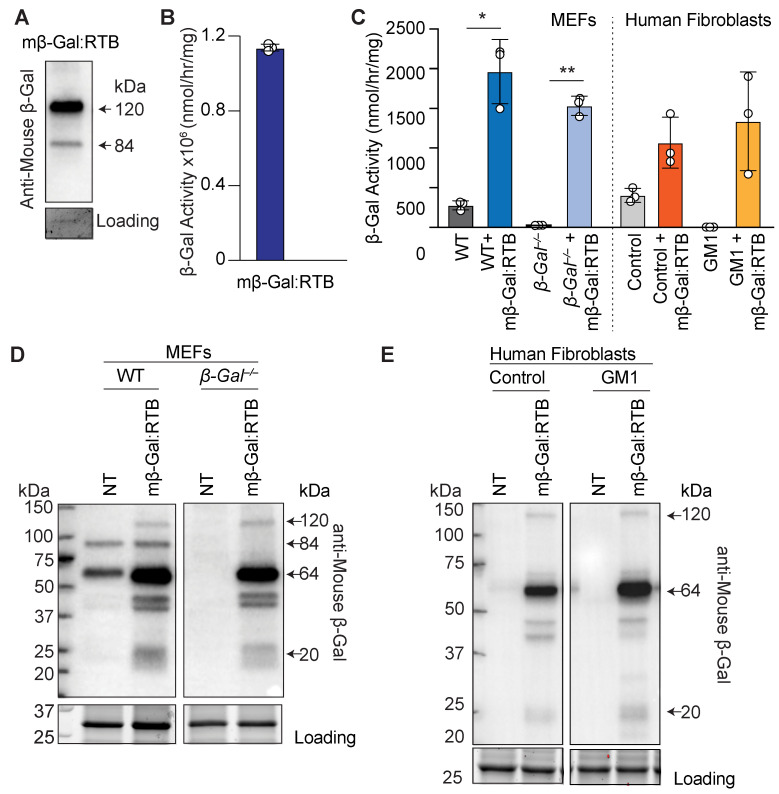
Cells treated with mβ-Gal:RTB show increased β-Gal activity and protein expression. (**A**) Immunoblot analyses of purified mβ-Gal:RTB probed with anti-mouse β-Gal antibody. Coomassie was used for loading control. (**B**) β-Gal activity of purified mβ-Gal:RTB. (**C**) β-Gal activity in MEFs and GM1 patients’ fibroblasts treated with mβ-Gal:RTB. *n* = 3. (**D**,**E**) Immunoblot analyses of MEFs (**D**) and human GM1 patients’ fibroblasts (**E**) treated with mβ-Gal:RTB and probed with anti-mouse antibody. NT = not treated. Data represents the means ± SD, * *p* < 0.05, ** *p* < 0.01.

**Figure 2 cells-11-02579-f002:**
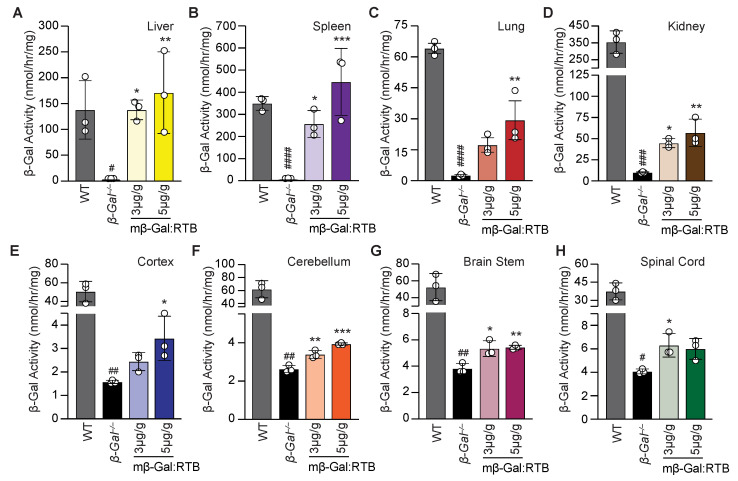
Dose-dependent treatment of β-Gal^−/−^ mice with mβ-Gal:RTB. (**A**–**D**) β-Gal activity of *β-Gal*^−/−^ mice treated with low (3 µg/g mouse) and high (5 µg/g mouse) dose mβ-Gal:RTB in the liver (**A**), spleen (**B**), lung (**C**), and kidney (**D**). *n* = 3. (**E**–**H**) β-Gal activity of *β-Gal*^−/−^ mice treated with low (3 µg/g mouse) and high (5 µg/g mouse) dose mβ-Gal:RTB in the different CNS regions: cortex (**E**), cerebellum (**F**), brain stem (**G**) and spinal cord (**H**). *n* = 3. Data represents the means ± SD, * indicates significance between untreated and mβ-Gal:RTB treated β-Gal^−/−^ mice, * *p* < 0.05, ** *p* < 0.01, *** *p* < 0.001. # indicates significance between WT and untreated β-Gal^−/−^ mice, # *p* < 0.05, ## *p* < 0.01, ### *p* < 0.001, #### *p* < 0.0001.

**Figure 3 cells-11-02579-f003:**
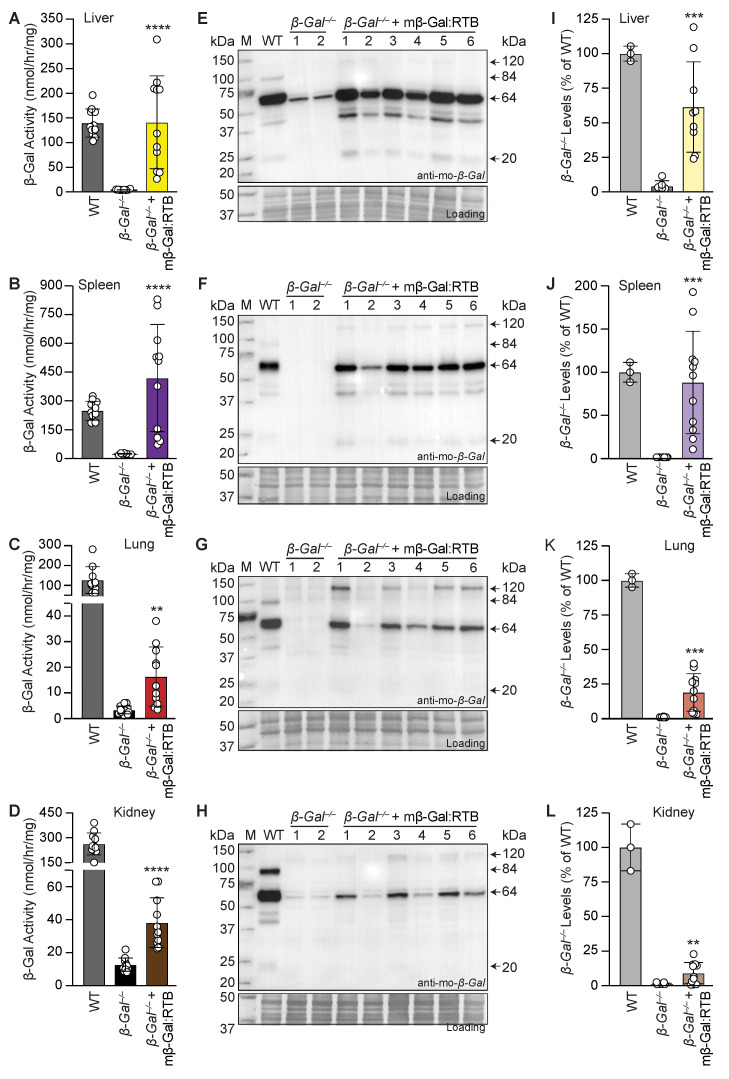
Long-term treatment with mβ-Gal:RTB shows increased β-Gal activity and protein levels in visceral organs. (**A**–**D**) β-Gal activity in liver (**A**), spleen (**B**), lung (**C**) and kidney (**D**) of *β-Gal*^−/−^ mice treated with mβ-Gal:RTB for 6 weeks. *n* = 11. (**E**–**L**) Immunoblots analyses and corresponding quantifications of *β-Gal*^−/−^ mice treated with mβ-Gal:RTB in the liver (**E**,**I**), spleen (**F**,**J**), lung (**G**,**K**) and kidney (**H**,**L**). WT: *n* = 3, *β-Gal*^−/−^: *n* = 5, *β-Gal*^−/−^ + mβ-Gal:RTB: *n* = 11. Data represents the means ± SD, ** *p* < 0.01, *** *p* < 0.001, **** *p* < 0.0001.

**Figure 4 cells-11-02579-f004:**
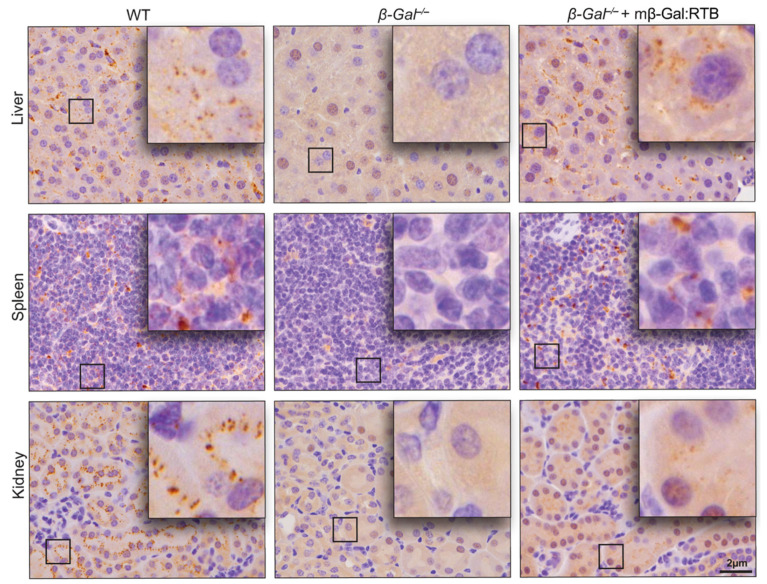
*β-Gal*^−/−^ mice treated with mβ-Gal:RTB show β-Gal puncta in visceral organs. IHC staining of the liver (**top** panel), spleen (**middle** panel) and kidney (**bottom** panel) of WT, *β-Gal*^−/−^ and injected *β-Gal*^−/−^ mice using anti-mouse β-Gal antibody. Inset: 4× magnification of black box. Scale bar: 20 µm.

**Figure 5 cells-11-02579-f005:**
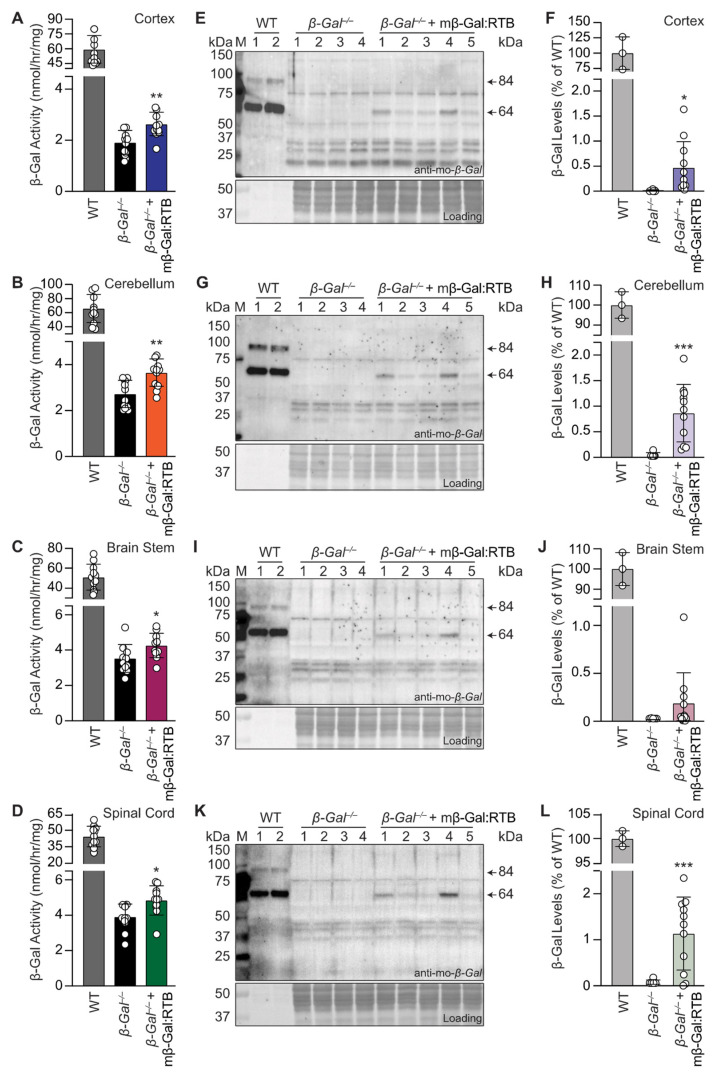
Long-term treatment with mβ-Gal:RTB shows increased β-Gal activity and protein levels in the CNS. (**A**–**D**) β-Gal activity in cortex (**A**), cerebellum (**B**), brain stem (**C**) and spinal cord (**D**) of *β-Gal*^−/−^ mice treated with mβ-Gal:RTB for 6 weeks. *n* = 11. (**E**–**L**) Immunoblots analyses and corresponding quantifications of *β-Gal*^−/−^ mice treated with mβ-Gal:RTB in the cortex (**E**,**F**), cerebellum (**G**,**H**), brain stem (**I**,**J**) and spinal cord (**K**,**L**). WT: *n* = 3, *β-Gal*^−/−^: *n* = 5, *β-Gal*^−/−^ + mβ-Gal:RTB: *n* = 11. Data represents the means ± SD, * *p* < 0.05, ** *p* < 0.01, *** *p* < 0.001.

**Figure 6 cells-11-02579-f006:**
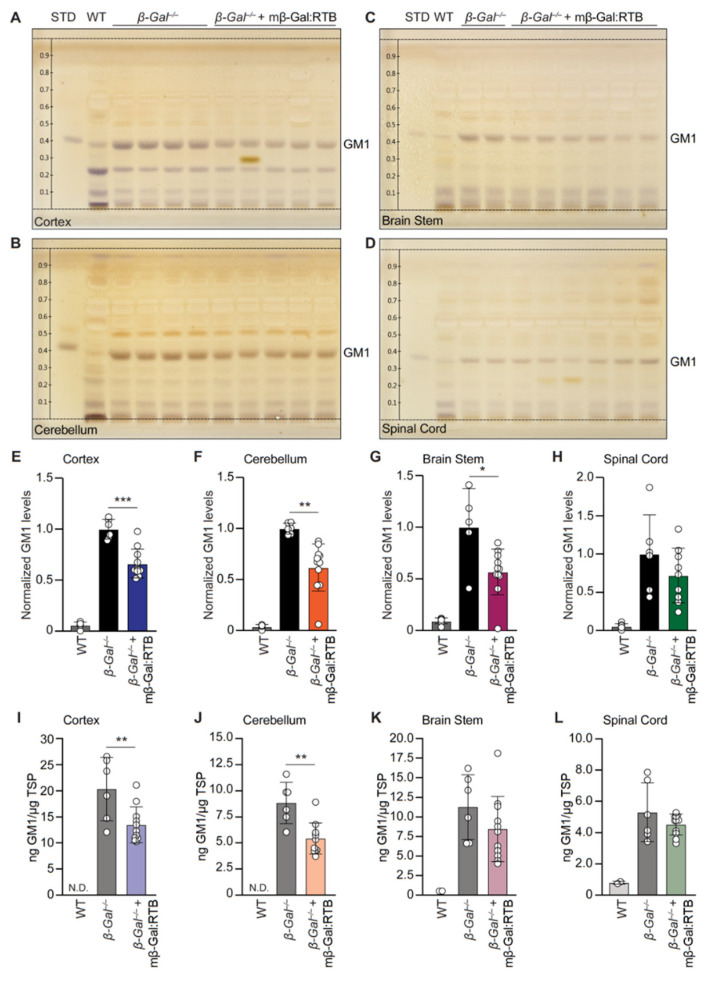
mβ-Gal:RTB treated mice show reduction of GM1 in the CNS. (**A**–**D**) HPTLC analyses of GM1 levels in the CNS regions of WT, *β-Gal*^−/−^ and treated mice. STD: GM1 standard (1 µg). (**E**–**H**) Quantification of GM1 levels in A. *n* = 11 (**I**–**L**) Quantification of GM1 levels measured by ELISA in the cortex (**I**), cerebellum (**J**), brain stem (**K**), and spinal cord (**L**). WT: *n* = 3, *β-Gal*^−/−^: *n* = 6, *β-Gal*^−/−^ + mβ-Gal:RTB: *n* = 11. Data represents the means ± SD, * *p* < 0.05, ** *p* < 0.01, *** *p* < 0.001.

**Figure 7 cells-11-02579-f007:**
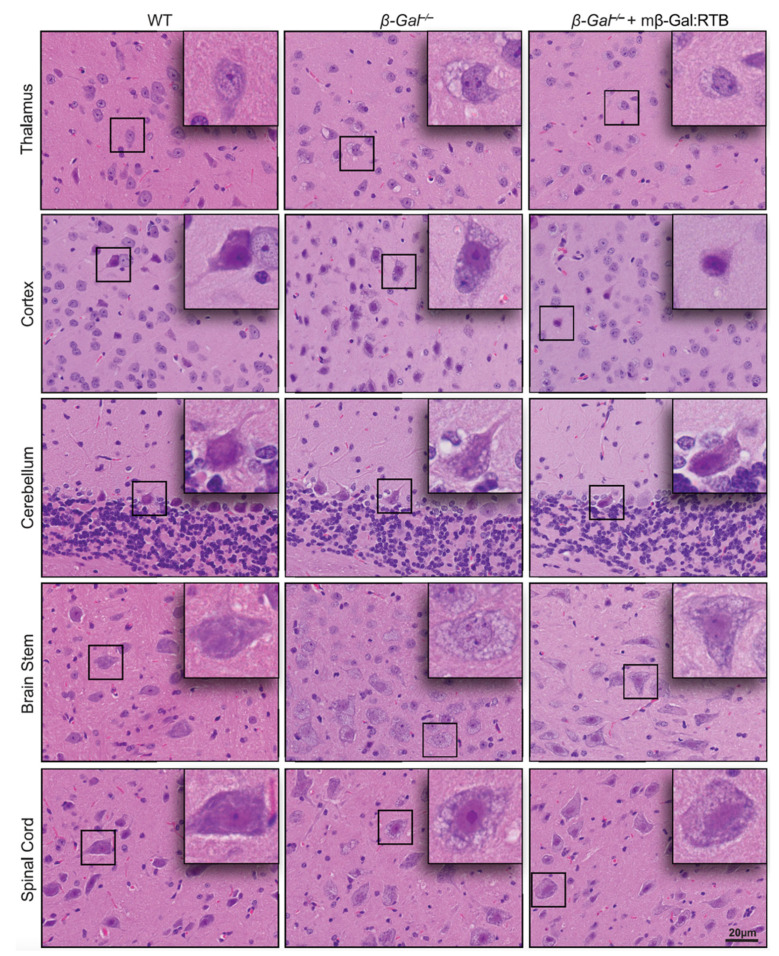
Reduced neuronal vacuolation in mβ-Gal:RTB treated mice. H&E staining of the CNS show reduced vacuolation in neurons of the thalamus, cortex, cerebellum, brain stem, and spinal cord of treated mice. Scale bar: 20 µm. Insets: 4× magnification of black boxes showing a single neuron.

**Figure 8 cells-11-02579-f008:**
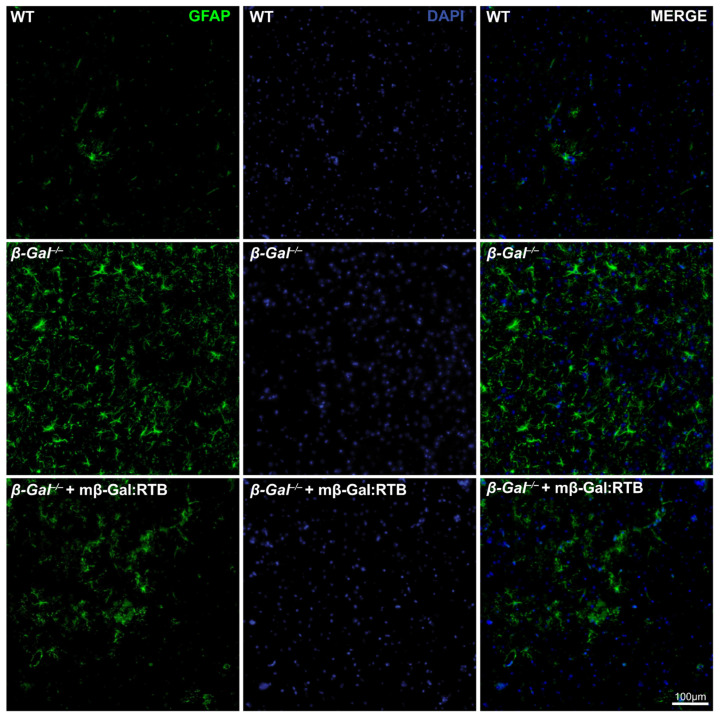
Reduced astrogliosis in mβ-Gal:RTB treated *β-Gal*^−/−^ mice. IF staining of astrocytes in the thalamic region of WT, *β-Gal*^−/−^ and mβ-Gal:RTB injected *β-Gal*^−/−^ mice using anti-GFAP antibody. Scale bar: 100 µm.

**Figure 9 cells-11-02579-f009:**
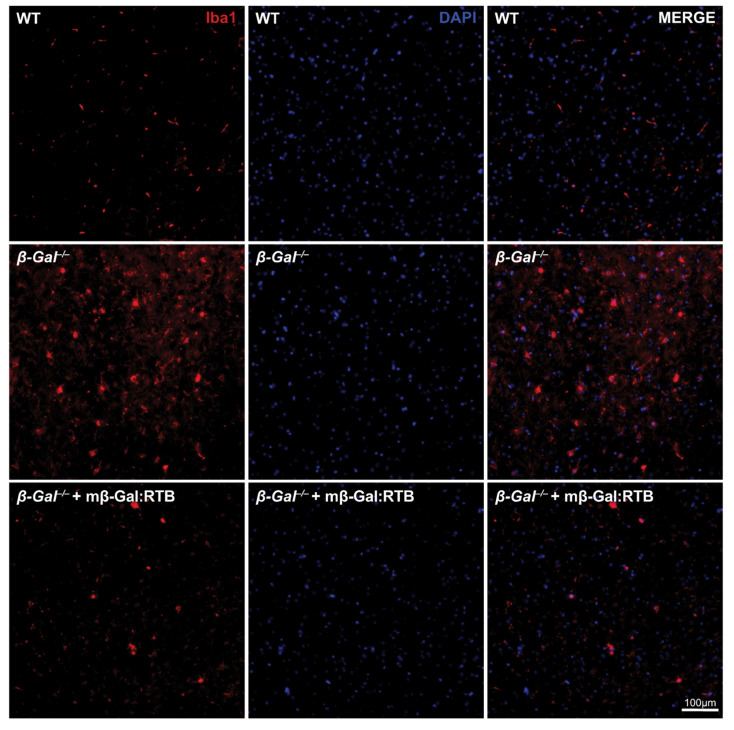
Reduced microgliosis in mβ-Gal:RTB treated *β-Gal*^−/−^ mice. IF staining of microglia in the thalamic region of WT, *β-Gal*^−/−^ and mβ-Gal:RTB injected *β-Gal*^−/−^ mice using anti-Iba1 antibody. Scale bar: 100 µm.

**Figure 10 cells-11-02579-f010:**
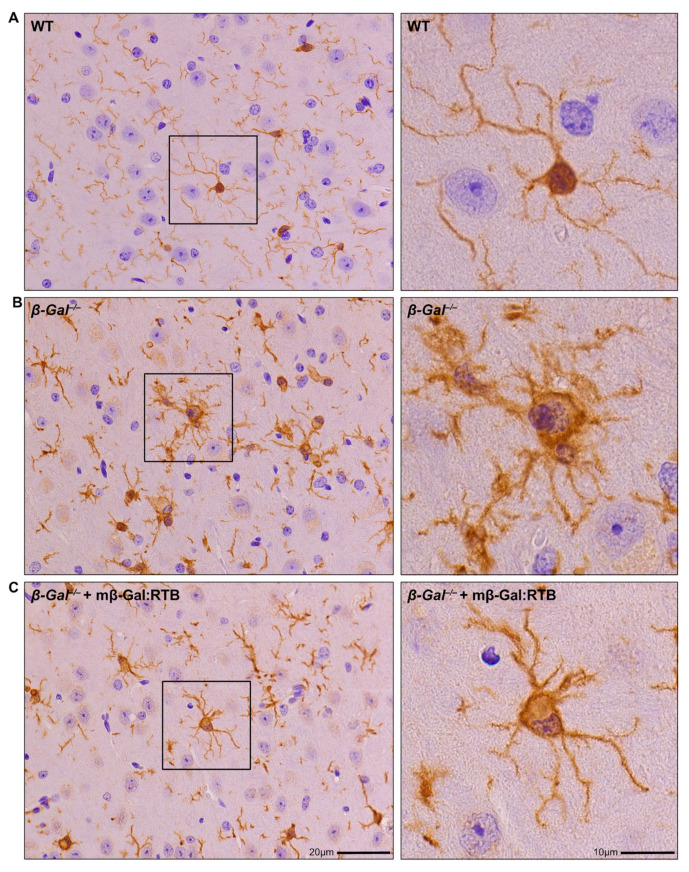
Microglia from mβ-Gal:RTB treated mice show reduced cell body size. (**A**–**C**) IHC analyses of the thalamic region of WT (**A**), *β-Gal*^−/−^ (**B**), and mβ-Gal:RTB injected (**C**) mice using anti-Iba1 antibody. Images on right show 3× zoom of black boxes in overview images on left. Scale bars: 20 and 10 µm.

## Data Availability

Not applicable.

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
