# Peer review of "Preclinical Enzyme Replacement Therapy with a Recombinant β-Galactosidase-Lectin Fusion for CNS Delivery and Treatment of GM1-Gangliosidosis"

_cells, 2022, doi:10.3390/cells11162579_

Round 1
Reviewer 1 Report
General comment:
The manuscript highlights a potentially less invasive approach to treat systemic and neurological disease associated with GM1 gangliosidosis.
There is compelling evidence here to suggest that an intravenous administered ERT approach with mbGal-:RTB can deliver Beta-Galactosidase activity to the brain in sufficient amounts to partially turnover gangliosides over a 6 week period of twice weekly dosing.
A few minor specific comments should be addressed in a revised manuscript.
Specific comments:
Comment 1:
Line 117: What is LCM?
Comment 2:
Section 2.3: While there are some details mentioned about expression, the method should include information on production scale and purification method for mBGal-RTB. Information around lot-to-lot variation should also be mentioned in the manuscript and whether this process requires further optimization.
Comment 3:
Could the authors comment on the mechanism of cellular uptake and processing of mbGal:RTB in the brain? Is this occurring in a RTB-dependent manner or Man6P-independent manner or both? In Fig 1 was cellular uptake of purified mB-Gal:RTB in MEFs, and / or fibroblasts partially inhibited with Man6P, or was uptake mediated entirely in a Man6P-independent manner? Competitive uptake studies in MEFs or primary neuronal / microglial / astrocytes in future in vitro studies could help to provide a mechanism for why a RTB-fusion ERT approach seems to be working to target the enzyme to the brain and partially turnover gangliosides.
Comment 4:
The authors showed convincingly that mBGal is reaching the lysosomal compartment by Western blotting in various tissues of treated mice. Did the authors determine the half-life of mbGal:RTB following cellular uptake and delivery to lysosomes in MEFs or fibroblasts? This could help to inform dosing frequency in future pre-clinical development.
Comment 5:
A limitation of the study is that Neu1 activity levels were not evaluated. Neu1 activity and Neu1 protein have previously been shown to be elevated in the brain tissue in the d'Azzo GLB1 -/- mouse model due to Beta-Gal being a negative regulator for Neu1 (PMID:33188082). Furthermore, elevated Neu1 activity in the GLB1 -/- brain can be normalized with ICV-ERT with rhBeta-Gal (PMID: 33188082). The advantage of an intermittent ERT-dosing approach is that Neu1 activity in the brain of GLB1 -/- mice can be normalized, and the dosing frequency of ERT can be adjusted to avoid secondary Neu1 deficiency. The authors should mention this need for optimizing dose-frequency and monitoring of Neu1 activity with mbGAL:RTB in future preclinical studies in the discussion.
Comment 6:
Why do the Beta gal -/- mice have B-gal activity present? Presumably this is something to do with the 4MU substrate detecting endogenous activity of other galactosidases. Please mention this in the results.
Comment 7:
Please mention in methods whether tissues were perfused with saline prior to collection of tissues for histology / biochemistry?
Comment 8:
Line 462: please change intraventricular to intracerebroventricular.
Comment 9:
Another potential limitation for this approach was the use or CPH to reduce the inflammatory response to the RTB-fusion enzyme. Please address in the discussion how this can potentially be overcome in further pre-clinical development studies for a RTB-fusion ERT approach for GM1 gangliosidosis patients. Presumably, PEGylation of the plant-produced enzyme can help to mitigate the immune response.
Reviewer 2 Report
This paper is very interesting. I have only one main point.
My concern is on the presentation of results related to GM1 content.
For a correct understanding of the GM1 decrease after treatment, I would like that a TLC chemically stained ganglioside pattern was presented for control animals, GM1 gangliosidosis animals and treated GM1 gangliosidosis animals.The total ganglioside content should be also reported.
Reviewer 3 Report
Very good piece of work. The strategy to produce from the plant the recombinant protein fused with the RTB protein demonstrates useful to cross the brain barriers. The protein is functioning and furthermore leads improvements in the used models. The work aims to demonstrate that the protein is present and functioning in the different visceral parts, although in different extent, and moreover that it can arrive at the brain, ameliorating the neurological deficiencies.
The work is very well done.
Some acronyms should be explained in the text.
Reviewer 4 Report
-
· Line 185 – spelling of florescent?
· Figure 2 - units for enzyme activity use “he” rather than “hr” - which is found in other figures.
· Figure 2 – presence of enzyme in brain is much lower than in other tissues? Why?
· Figure 3 – Can the authors explain the wide variability in the enzyme activity levels in each tissue type. For example in Figure 2 both the 3 and 5 ug/g doses in lung show no overlap with untreated lysates. However, in figure 3 there is overlap between affected and treated?
· In figure 3E, bands are noted for beta gal in the -/- tissue? This is not seen, at least to this degree, in other tissues?
· For figure 6 can the author explain why some data points within each tissue type overlap with WT and others show a decrease?
· Can the GM1 storage material be found in other tissues besides the brain? If so, did the authors look for a reduction in GM1 in the tissues with higher enzyme activity levels.
· Since the B-gal enzyme also binds to the NEU1 protein, did the authors assess NEU1 activity?
· Is it possible that there is an immune response to the fusion protein which is red
· I have difficulty understanding which samples were used for the data in the blots in figure 3. Are these representative of a subset? The figure legend says that more samples were used than were shown?
· Given the authors have a rough quantitation of the amount of enzyme present in each sample (blots), can the authors make a specific activity measurement – beta-gal enzyme / amount of enzyme present?
· Comment: The authors sacrificed the animals at 10 weeks (2.5 months). According to Eikelberg et al the clinical symptoms showed up at 3.5 months. Did the authors continue to treat animals until clinical symptoms typically appear?
· Comment: Several methods exist to measure GM1 and the oligosaccharides that accumulate in GM1 patients using mass spectrometry. These methods may provide a wider dynamic range to assess storage.
Round 2
Reviewer 1 Report
My concerns have been addressed in this in this revised version of the manuscript. I would recommend publishing. Congrats to the authors for a great contribution toward developing an effective therapy for GM1 gangliosidosis.Author Response
My concerns have been addressed in this in this revised version of the manuscript. I would recommend publishing. Congrats to the authors for a great contribution toward developing an effective therapy for GM1 gangliosidosis.
R: We thank the Reviewer for these comments.
Reviewer 2 Report
I would prefer that the original tlc plates was presented in the article, not in the supplementary files
Author Response
I would prefer that the original tlc plates was presented in the article, not in the supplementary files.
R: We have exchanged the images in Figure 6A for full TLC plate images, which are now Figure 6A-D, and have adjusted the legend and text accordingly.